# Identification of a Major QTL-Controlling Resistance to the Subtropical Race 4 of *Fusarium oxysporum* f. sp. *cubense* in *Musa acuminata* ssp. *malaccensis*

**DOI:** 10.3390/pathogens12020289

**Published:** 2023-02-09

**Authors:** Andrew Chen, Jiaman Sun, Guillaume Martin, Lesley-Ann Gray, Eva Hřibová, Pavla Christelová, Nabila Yahiaoui, Steve Rounsley, Rebecca Lyons, Jacqueline Batley, Ning Chen, Sharon Hamill, Subash K. Rai, Lachlan Coin, Brigitte Uwimana, Angelique D’Hont, Jaroslav Doležel, David Edwards, Rony Swennen, Elizabeth A. B. Aitken

**Affiliations:** 1School of Agriculture and Food Science, The University of Queensland, Brisbane, QLD 4067, Australia; 2School of Life Science, Jiaying University, Meizhou 514015, China; 3CIRAD, UMR AGAP Institut, F-34398 Montpellier, France; 4UMR AGAP Institut, Univ Montpellier, CIRAD, INRAE, Institut Agro, F-34398 Montpellier, France; 5Australian Genome Research Facility, Victorian Comprehensive Cancer Centre, Melbourne, VIC 3000, Australia; 6Centre of the Region Haná for Biotechnological and Agricultural Research, Institute of Experimental Botany of the Czech Academy of Sciences, 77200 Olomouc, Czech Republic; 7Inari Agriculture, West Lafayette, IN 47906, USA; 8School of Biological Sciences, The University of Western Australia, Perth, WA 6907, Australia; 9Department of Agriculture and Fisheries, Maroochy Research Facility, Nambour, QLD 4560, Australia; 10Genome Innovation Hub, University of Queensland, Brisbane, QLD 4072, Australia; 11Department of Microbiology and Immunology, Peter Doherty Institute for Infection and Immunity, University of Melbourne, Melbourne, VIC 3004, Australia; 12International Institute of Tropical Agriculture, Kampala P.O. Box 7878, Uganda; 13Laboratory of Tropical Crop Improvement, Division of Crop Biotechnics, Katholieke Universiteit Leuven, 3001 Leuven, Belgium

**Keywords:** banana, fusarium wilt, host resistance, quantitative trait locus, bulk segregant analysis, *Fusarium oxysporum* f. sp. *cubense*, Subtropical Race 4, QTL-seq

## Abstract

Vascular wilt caused by the ascomycete fungal pathogen *Fusarium oxysporum* f. sp. *cubense* (*Foc*) is a major constraint of banana production around the world. The virulent race, namely Tropical Race 4, can infect all Cavendish-type banana plants and is now widespread across the globe, causing devastating losses to global banana production. In this study, we characterized *Foc* Subtropical Race 4 (STR4) resistance in a wild banana relative which, through estimated genome size and ancestry analysis, was confirmed to be *Musa acuminata* ssp. *malaccensis*. Using a self-derived F_2_ population segregating for STR4 resistance, quantitative trait loci sequencing (QTL-seq) was performed on bulks consisting of resistant and susceptible individuals. Changes in SNP index between the bulks revealed a major QTL located on the distal end of the long arm of chromosome 3. Multiple resistance genes are present in this region. Identification of chromosome regions conferring resistance to *Foc* can facilitate marker assisted selection in breeding programs and paves the way towards identifying genes underpinning resistance.

## 1. Introduction

Banana, as a fruit or subsistence crop, provides sources of essential nutrients in daily dietary uptake for millions of people around the world [1]. The fungal vascular wilt disease of banana, often referred to as Panama disease, has put major constraints on global banana production. The causal agent underlying this disease is the ascomycete fungus *F. oxysporum* f. sp. *cubense* (*Foc*). *Foc* is a highly adaptive pathogen, that is composed of different evolutionary lineages, currently represented by 24 vegetative compatibility groups (VCGs) [2,3,4]; alternatively, it can be classified into a race structure based on the range of host cultivars that are affected [5]. *Foc* race 1 decimated banana production primarily based on the triploid cultivar ‘Gros Michel’ in the mid-20th Century. Its replacement, ‘Cavendish’, is resistant to race 1 and has since dominated the global market. Today, global banana production has succumbed to the tropical race 4 (TR4) of *Foc*, following its spread and detection in Asia and Pacific region in the 1980–1990s [6,7], that is virulent on all ‘Cavendish’ type banana. The pandemic that followed has once again put global banana production under siege.

*Foc* is a soil borne pathogen that enters the host plant through roots where it travels through the plant’s vasculature to colonize the entire plant [8,9]. Proliferation of the fungus leads to blockages in the water conducting vessels of the xylem, thereby stopping water and nutrient supplies to the plant [10]. This eventually leads to wilting and death of the plant. Once *Foc* establishes itself in the soil, it can remain in the soil for decades [10,11,12], surviving in the form of thick-walled survival spores (chlamydospores) on plant debris, or as an endophyte on alternative weed hosts like *Chloris inflata* (common grass), *Amaranthus* and *Paspalum* spp. [13,14,15]. *Foc* is dispersed through the movement of soil, water, infected plant material, and animals [5]. Despite best management practices and biosecurity measures being put in place to contain, control, eradicate, or exclude *Foc* from infected plantations, *Foc*-TR4 has continued to spread around the world, with potential to occupy 17% of banana cultivated lands over the next two decades, projecting losses up to 36 million tons of production worth over USD 10 billion [16].

Genetic resistance to *Foc* will lead to long-term solutions for the management of *Fusarium* wilt [17,18]. Research activities in developing *Foc* resistant cultivars through either conventional breeding or transgenic approaches are predicted to have significant impact in the reduction of potential losses caused by this disease [19]. Currently, no commercial cultivars are available that possess complete resistance against *Foc*-TR4 [5]. Wild diploid relatives of banana are known to harbor *Foc*-TR4 and subtropical race 4 (STR4) resistance [20,21]. In Australia, TR4 is a strictly quarantined matter, and due to logistic and experimental constraints, STR4 was used in this study instead of TR4. STR4 strains are only known to affect ‘Cavendish’ banana in the subtropics, whereas TR4 strains are known to cause serious impacts in both tropics and subtropics [22]. Here, we report the characterization of STR4 resistance in a wild banana relative, which we confirmed to be of *Musa acuminata* ssp. *malaccensis* origin. Genome sequencing and SNP resistance association analysis using an F_2_ derived population were performed to detect quantitative trait loci (QTL) controlling resistance against *Foc*-STR4. Potential candidate genes were identified. This study paves the way for molecular assisted breeding tools to be developed in the selection of resistant genotypes to different forms of *Foc*.

## 2. Materials and Methods

### 2.1. Population

Banana tissue culture clones from three *Foc*-TR4 resistant (R) lines (‘Ma850’, ‘Ma851’, ‘Ma852’) and three *Foc*-TR4 susceptible (S) lines (‘Ma845’, ‘Ma846’, ‘Ma848’) were micro-propagated and maintained in vitro as described in a previous study [23]. The R lines are self-derived progeny from a single wild progenitor. The S lines are self-derived progeny from another wild progenitor, not related to the R progenitor. Single gene resistance (3R:1S) to *Foc*-STR4 (VCGs 0120, 0129, 01211) and *Foc*-TR4 (VCG 01213/16) has been previously detected in the progeny of the R lines in pot trials conducted at the University of Queensland (QLD), and the Coastal Plains Research Facility (NT) [21]. An F_2_ population carrying resistance was developed using three progeny crosses/selfs, comprising ‘Ma851’ × ‘Ma851’, ‘Ma851’ × ‘Ma852’, and ‘Ma852’ × ‘Ma852’. An initial set of proliferative F_2_ lines from these crosses were multiplied in vitro and then challenged with three combined distinct isolates of *Foc*-STR4 VCG0120 in a pot trial with plants grown under ambient (25–28 °C/day and 18–20 °C/night) conditions. A previously described millet *(Echinochloa esculenta*) inoculation technique was used where plants, 12 weeks post tissue culture, were challenged in glasshouse conditions with *Foc* inoculum and scored for internal symptoms after another 12 weeks [9]. Based on the resultant phenotype, 27 *Foc*-STR4 resistant and 25 susceptible progenies were selected for a whole genome-wide QTL sequencing (QTL-seq) analysis.

### 2.2. Next Generation Sequencing

DNA extraction was performed using a phenol–chloroform method [24] on tissue from young banana leaves. DNA was cleaned up using Ampure XP beads (Beckman Coulter, Brea, CA, USA) according to manufacturer’s instructions, and then pooled at equal molar ratios to form two bulks (R and S). The bulk DNA, along with the DNA of parental lines ‘Ma851’ and ‘Ma848’ were sent to AGRF (Australian Genome Research Facility, Brisbane, Australia) for Illumina library preparation (TrueSeq DNA Nano kit, Illumina, San Diego, CA, USA) and sequencing on a Hiseq4000 platform to generate 150 bp of paired-end reads. In total, 173 M, 197 M, 427 M, and 494 M reads were generated for ‘Ma848’, ‘Ma851’, R-bulk, and S-bulk, respectively.

### 2.3. QTL-Seq Analysis

QTL-seq analysis was performed by using a QTL-seq pipeline [25]—which includes adaptor trimming by ‘Trimmomatic’ [26]—alignment to the reference genome DH-Pahang v4.3 ‘https://banana-genome-hub.southgreen.fr/’ (accessed on 8 February 2023) by using ‘BWA-MEM’ [27], and calculation of changes in SNP indices between the R- and S-bulks. SNP index was calculated as count of SNP base/count of reads aligned. SNPs were filtered using minimum read depth of 8 and SNP index < 0.3. ΔSNP index was calculated by subtracting the SNP index of the S-bulk from that of the R-bulk. Plots were visualized with a window size of 2 Mb and a fixed step size of 100 Kb. ‘SnpEff’ v5 was run using default settings and the reference genome DH-Pahang v4.3 to calculate the number of variants per chromosome and the type of effect associated with these variants [28]. SNP and insertion/deletion variants were annotated by running ‘ANNOVAR’ with default settings against DH-Pahang v4.3 [29].

### 2.4. GO Enrichment Analysis

Go enrichment analysis was performed using the online tool at ‘https://banana-genome-hub.southgreen.fr/content/go-enrichment/’ (accessed on 8 February 2023). The running settings included p and q value cutoffs of 0.05, and 0.1, respectively. The input of the candidate region was defined from 36,275,176 bp to 42,483,366 bp position on chromosome 3 of ‘DH-Pahang’ v4.3. It contained 861 gene models and 638 associated GO terms.

### 2.5. Ancestry mosaic Analysis

The susceptible *M. acuminata* ssp. *malaccensis* ‘Ma848’ and a resistant progeny of ‘Ma851’ were sequenced using one lane of Illumina Hiseq4000 which produced 52.4 Gb and 59.6 Gb of data from the respective libraries sequenced. Variant calling, SNP clustering and ancestry analysis were performed using the methodology and the six banana ancestral groups described in a previous study [30].

### 2.6. Simple Sequence Repeat (SSR) Genotyping

SSR genotyping was performed on the six *M. acuminata* ssp. *malaccensis* parents (‘Ma845’, ‘Ma846’, ‘Ma848’, ‘Ma850’, ‘Ma851’, ‘Ma852’) using an existing pipeline [31]. Nineteen SSR loci were amplified using a set of M13 tailed fluorescent labelled primers. Estimation of allele sizes on an ABI 3730xl DNA analyzer (Applied Biosystems, Waltham, MA, USA), and the analysis of the data using GeneMarker v1.75 (Softgenetics, PA, USA) were performed [32]. The marker data was then analyzed together with a set of *Musa* accessions that has been verified with SSR genotyping. The core set includes East African Highland Banana (EAHB), *Musa acuminata* ssp. (AA), *M. balbisiana* (BB), and hybrids containing AB, AAB and ABB genomes, as well as *M. schizoparpa* and AS hybrids, and representatives of other sections of *Musa*, namely *Rhodochlamys*, *Australimusa*, and *Callimusa* [31,32]. Calculation of genetic distances among individual accessions and hierarchical clustering of the distance matrix using unweighted pair group method with arithmetic mean (UPGMA) were performed using the settings, as described previously [32].

### 2.7. Nuclear Genome Size Estimation

Suspensions of intact cell nuclei of ‘Ma848’ and ‘Ma851’ were prepared from fresh leaf “cigars”. Nuclear DNA was stained by DAPI (4′,6-diamidino-2-phenylindole) and relative fluorescence intensity of stained nuclei was analyzed using Partec PAS flow cytometer (Partec, Münster, Germany) equipped with a high-pressure mercury lamp as excitation light source as per a previous study [33]. Chicken red blood cell nuclei (CRBC) were included in the samples and served as an internal reference standard.

## 3. Results

Flow cytometric analyses produced histograms of relative nuclear DNA content (Figure 1A), comprising two dominant peaks representing G_1_ nuclei of ‘Ma848’, ‘Ma851’, and chicken red blood cell nuclei (CRBC). A peak ratio (‘Ma848’ or ‘Ma851’: CRBC) in the range of 0.51 to 0.53 indicates that both ‘Ma848’ and ‘Ma851’ carry diploid genomes.

Nineteen SSR loci were amplified in all six ‘Ma’ accessions used in this study to add to the published *Musa* UPGMA dendrogram [32]. Thirteen clusters were generated, representing the core collection of *Musa* sp. [32,33]. Wild A-genome progenitors were grouped together with related diploids and triploid accessions and are generally in agreement to the morphological traits-based classification of groups [33]. ‘Ma845’, ‘Ma846’, ‘Ma848’, ‘Ma850’, ‘Ma851’, ‘Ma852’ were grouped together with *M. acuminata* ssp. *malaccensis* accessions, and diploid AA cultivars from the ‘Island South-East Asia’ (ISEA) region in cluster VI (Figure 1B). Clustering of these lines within the *M. acuminata* ssp. *malaccensis* subgroup show that they are most closely related to *M. acuminata* ssp. *malaccensis* ‘Pa Songkhla’, ‘Kluai Pal’, and AA cv. ‘Pisang Sintok’ (Figure 1B, Appendix A).

Six ancestries of diploid origin were previously identified [30]. Informative alleles representing the six ancestral groups were selected based on a correspondence analysis of 13 accessions and then used to assign contiguous regions of the same ancestral group to ‘Ma848’ and ‘Ma851’. The clustered alleles represent six groups of origin, namely *Musa balbisiana*, *M. acuminata ssp. burmannica*/*siamea*, *M. acuminata* ssp. *malaccensis*, *M. acuminata* ssp. *banksia*/*microcarpa*, *M. acuminata* ssp. *zebrina*, and ‘AA’ cv. ‘Pisang Madu’. Statistical assessment of the expected allele frequency showed that *M. acuminata* ssp. *malaccensis* (group 3) was the predominant ancestry assigned to chromosomes in ‘Ma848’ and ‘Ma851’ (Figure 1C). ‘Ma851’, which carries *Foc*-STR4 and TR4 resistance, showed a *M. acuminata* ssp. *malaccensis* constitution with apparently limited introgressions from another ancestry. ‘Ma848’ showed a similar composition, but with large regions of ancestry other than *M. acuminata* ssp. *malaccensis*, namely *M. acuminata* ssp. *banksia/microcarpa* (group 4) and ‘AA’ cv. ‘Pisang Madu’ (group 5), observed on chromosome groups 3, 5 and 9 (Figure 1C). Both accessions also showed a high level of heterozygosity as most regions on chromosome 3 only had a single haplotype called. Overall, the local ancestry predictions on both lines suggest that they are both of *M. acuminata* ssp. *malaccensis* origin.

QTL-seq was applied to detect QTL involved in *Foc*-STR4 resistance in an F_2_ population derived from self-crossed ‘Ma851’ and ‘Ma852’. Both lines are resistant to *Foc*-STR4 and their F_2_ progeny segregated for *Foc*-STR4 resistance at a resistant and susceptible ratio of approximately 3 to 1 (Chi-square goodness of fit with χ^2^ = 0.056, *p* = 0.81, *df* = 1, *α* = 0.05). Sequencing of the R- and S- bulks and mapping of these reads to the reference genome produced a total of 3.47 million variants, across all 11 chromosomes and the mitochondria (Table 1). The reference genome ‘DH-Pahang’ is derived from the *M. acuminata* ssp. *malaccensis* accession ‘CIRAD930’ (ITC1511) and is resistant to *Foc*-TR4 [34]. It is also closely related to our lines (Appendix A). The majority of the SNPs are positioned in the non-coding and intronic regions, with only 21,178 (0.3%) and 263,738 (3.7%) variants occurring in splicing sites and exons, respectively (Appendix A). Sequencing data from the R- and S-bulks were used to calculate the SNP index, which is the proportion of short reads (*k*) harboring SNPs that are different from the reference, covering a particular genomic position [25]. An SNP index of 0.5 indicates that each bulk contributes equally to the variation. The Δ(SNP index) was calculated for a given genomic interval in a 2 Mb sliding window analysis, which detected a region of significant effect on chromosome 3, with statistical confidence intervals (CIs) of *p* < 0.05 and *p* < 0.01 under the null hypothesis of no QTL present in these regions (Figure 2). A 6.3 Mb region was significantly associated with the *Foc*-STR4 resistance in the 99% CI at the distal end of chromosome 3, from 36.2 to 42.5 Mb. A total of 861 genes are annotated in this region of DH-Pahang v4.3. Out of these, 75 annotated genes may have roles in disease resistance response, including 27 putative receptor-like kinases (RLK), 17 putative receptor-like protein (RLP), 28 putative nucleotide binding site and leucine rich repeat (NBS-LRR) proteins, and 1 homolog of the non-expressor of pathogenesis-related genes 1 (NPR1) (Appendix A). GO enrichment using the 861 candidate genes detected a single significant term, GO:0043531 (ADP binding), under molecular function, with *p* = 1.72 × 10^−23^ and a gene ratio of 33/638. The enrichment is associated with the 28 putatively defined NBS-LRR genes in the candidate region (Appendix A). No other enriched GO terms were detected under the other categories. The genomic position exhibiting the highest Δ(SNP index) is at 40.9 Mb position on chromosome 3 and has a mean Δ(SNP index) of 0.81. No regions of significant associations were detected on any other chromosomes (Figure 2).

## 4. Discussion

Fusarium wilt, also known as Panama disease, is a devastating disease that affects banana plants. Epidemics owing to Fusarium wilt have put major constraints on global banana production both historically and at the present time. So far, *Foc*-TR4 has significantly curtailed banana production in Australia, China, Indonesia, Malaysia, and the Philippines [6,7] and has spread to locations as far as Mozambique in Africa [35] and Colombia and Peru in South America [36,37]. The disease is posing a major threat to banana production, limiting the selection of cultivars and the land suitable for commercial production, while at the same time, putting constraints on food security of smallholders. Genetic resistance to *Foc* provides a long-term solution to the management of the disease. The identification of resistance is a step forward towards the development of *Foc*-TR4 resistant cultivars, either by using marker assisted selection or a transgenic approach. The diploid subspecies of *M. acuminata*, including *M. acuminata* ssp. *malaccensis*, *M. acuminata* ssp. *burmannica*, *M. acuminata* ssp. *microcarpa* and *M. acuminata* ssp. *siamea* are known to harbour *Foc* resistance [9,20,38].

In this study, we used high throughput SSR genotyping and flow cytometry as germplasm discovery tools to confirm the phylogroup and ploidy levels of a set of wild banana relatives, namely *M. acuminata* ssp. *malaccensis* within the *Musa* collection. Molecular characterization of wild *Musa* accessions such as this was only possible with the availability of germplasm collections, as well as the accompanying genetic and morphological data. The *Musa* Germplasm Information system (MGIS, ‘http://www.crop-diversity.org/banana/’ (accessed on 8 February 2023)), maintained by Bioversity International, provides the framework for the *Musa* collection to be categorized and characterized. Furthermore, the current system of SSR genotyping provides a reliable phylogenetic classification of wild relatives and hybrids that can be integrated into the core *Musa* collection to facilitate banana research and breeding, and at the same time, improve the management and conservation of global *Musa* germplasm collections [33].

Genome resequencing can be employed to mine SNP data to characterize evolutionary origins of genome segments in *Musa* species [30,39]. This is especially relevant in banana, as the hybridization between *Musa* species and subspecies is associated with the origin of cultivated banana [40,41]. *M. acuminata* subspecies are distributed along Southeast Asia and western Melanesia [41,42,43,44]. Hybridizations between some of them and with other *Musa* sp. gave rise to diploids and triploids selected for fruit edibility, leading to the diversity we have now in banana cultivars. With the level of genome diversity observed in *Musa*, determination of ancestral origin becomes important as it reveals genome mosaics or components that might be underlying important traits [30]. ‘Ma848’ and ‘Ma851’ seem to be derived from generally pure *M. acuminata* ssp. *malaccensis* sources. The presence of some segments of different origin, as clearly seen in ‘Ma848’, may have resulted from gene flow from partially fertile cultivars or from other wild individuals. The genome mosaic analysis provided a comprehensive view of the genome composition of the wild relatives and will be used in dissecting the different ancestral genetic pools present in some of the intraspecific hybrids or cultivars arising from these wild lines.

With genome sequencing becoming increasingly affordable, sizable reference genomes such as the *M. acuminata* ssp. *malaccensis* genome ‘DH-Pahang’ can be sequenced in high contiguity with long read technologies [45]. The latest version (v4) of this assembly contains entire chromosomes reconstructed in single contigs and serves as a valuable resource for dissecting genome regions of a high complexity, such as centromeres or clusters of paralogs. By using this reference, we performed a genome-wide QTL-seq analysis to detect a QTL region on chromosome 3 underpinning STR4 resistance. QTL for resistance against race 1 and TR4 has been previously detected on chromosome 10 [46].

Plant resistance to microbes is often mediated in a host–pathogen-specific manner, through the interactions between the products of a *R* resistance gene and corresponding *Avr* (avirulence) gene in the pathogen, more commonly now referred to as an effector [47]. R proteins have been differentiated on the basis of whether they are cytoplasmic or membrane-bound and can be further divided into classes on the presence of specific protein domains [48]. The STR4 resistance locus contains at least three groups of RLKs containing different ectodomains. These include a LRR ectodomain in kinases such as the GASSHO1 (GSO1, Macma4_03_g31320.1), a cysteine-rich galacturonan-binding ectodomain in the leaf rust 10 disease–resistance locus receptor-like protein kinase-like proteins (LRK10L); and two Gnk2 domains (the domains of unknown function 26, DUF) in the cysteine-rich protein kinases (CRKs). These RLKs belong to different classes within the RLK family, as determined by their structure [49,50]. Members of the RLK family play important roles in plant immunity, development, ABA signaling, and drought resistance [51,52,53,54,55]. The QTL region also harbors a cluster of extracellular LRR containing receptor-like proteins (LRR-LRPs) lacking a kinase domain, akin to receptors that have been shown to recognize xylanases from the *Trichoderma* species [56]; Ve1 and Ve2 receptors that provide race-specific resistance against *Verticillium* sp. [57]; and the Cf receptors providing resistance against *Cladosporium fulvum* in tomato [48].

Cytoplasmic R proteins are also predicted in this region, including NBS-LRRs that have an N-terminal coiled-coil (CC-NBS-LRR), akin to RPP13 that provides resistance to *Peronospora parasitica* in Arabidopsis [58]. Other NBS-LRRs belong to a group of *R* genes known as the resistance gene analogs (RGAs), isolated from *M. acuminata* ssp. *malaccensis* that lack the TIR motif but has either a CC or no obvious motif at the amino terminus [59,60]. RGA2, which is a CC-NBS-LRR protein, similar to I2 and Fom-2 [61,62], appears to be important in mediating *Foc*-R1 and *Foc*-TR4 resistance [63,64]. Furthermore, we have the non-expressor of pathogenesis-related genes 1 (NPR1), which has significant roles in establishing systemic acquired resistance (SAR) and induced systemic resistance (ISR) [65]. Lastly, Macma4_03_g24790.1 is similar to the Arabidopsis adenylyl cyclase AtLRRAC1 (At3g14460). AtLRRAC1 catalyzes the formation of the second messenger cAMP from ATP. A T-DNA insertion knock-out mutant atlrrac1-1 has enhanced susceptibility against *Golovinomyces orontii* and *Pseudomonas syringae*, suggesting a role of cAMP-dependent pathways in the defense biotrophic and hemibiotrophic plant pathogens [66].

The fact that the QTL underlies a complex region with duplicated R paralogs reflects the paleopolyploid nature of the banana genome. Three whole genome duplication events have been inferred throughout the history of the banana haploid genome [34,67,68], leading to an estimated one-third of the genes in multiple copies. Evolution by gene duplication is believed to be important for the gain and divergence of functions that may provide an evolutionary advantage [69]. Members of RLK family often have undergone expansions. It was found that more than 33% of RLK members are located in tandem clusters in Arabidopsis [70]. These expansions occur because of tandem duplications and whole-genome duplications. In Arabidopsis, tandem and proximal duplicates showed divergent functional roles but shared enriched GO terms critical for plant self-defense and adaptation, particularly in programmed cell death, immune response, and signaling receptor activity [71]. Tandem duplicates were characteristically enriched in GO terms involved in cofactor binding and enzymatic activities [71]. In this study, GO enrichment analysis on the candidate region revealed a single significant GO term (GO:0043531, ADP binding) under the GO category of molecular function, and identified an enrichment in the NBS-LRR gene cluster, consisting of the CC-NBS-LRR, RPP13-like protein 1, and RGA1-4 type resistance genes (Appendix A). This suggests that the NBS-LRR candidate genes are potential targets to investigate for the molecular dissection of resistance at this locus.

Next generation sequencing, when combined with bulked segregant analysis, offers rapid trait mapping at a high resolution [72,73]. Traditional bulked segregant analysis is time consuming [74] and is further complicated by the long turnover time of banana growth and constraints associated with phenotyping traits [75]. Bulked sample analysis and sequencing-based trait mapping have been described in many crop species [76,77]. Future work is currently being undertaken to fine map the QTL. Cleaved and amplified polymorphic sequence (CAPS) markers are being developed to saturate the candidate region. A linkage map is being developed. F_2_ individuals carrying recombination events are being identified. Recombinants will be phenotyped with both *Foc*-STR4 and *Foc*-TR4 to validate and delimit the candidate region. Potential functional SNPs in *R* gene candidates will be converted into markers to test for cosegregation with the race 4 resistances controlled by this locus. Comparative analysis of gene content in the QTL region through synteny analysis in the other resistant and susceptible *Musa* genomes can also lead to the identification of candidate resistance genes. At the same time, SNPs associated with *Foc* race 4 resistances can accelerate the development of *Foc* resistant cultivars through marker assisted selections in banana breeding programs.

## Figures and Tables

**Figure 1 pathogens-12-00289-f001:**
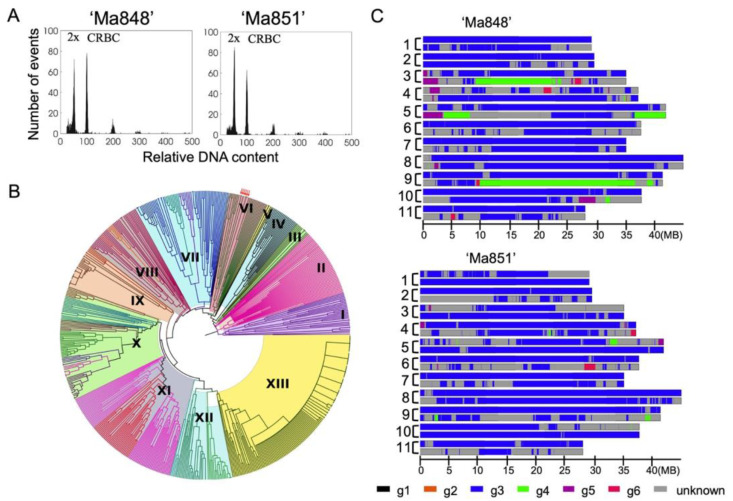
(**A**) Estimation of ploidy level of ‘Ma848’ and ‘Ma851’. Histograms of relative nuclear DNA content obtained through flow cytometric analysis. The internal reference was chicken red blood cell nuclei (CRBC) which was adjusted to appear at channel 100. Peaks at 200, 300, 400, 500 correspond to doublets, triplets and so on of the CRBC nuclei. G1 peak positions of *Musa* (‘Ma848’, ‘Ma851’): CRBC are 0.51 and 0.54, respectively, corresponding to a diploid genome. (**B**) A UPGMA dendrogram constructed with SSR data of *Musa* accessions including the six diploid lines from this study. Group **I**, *Rhodochlamys*; **II**, *Australimusa*, *Callimusa*; **III**, AA group: *burmannicoides*, *burmannica*, *siamea*; **IV**, *schizocarpa* (SS) and AS hybrids; **V**, AA group, *zebrina*, *truncata*; **VI**, AA group, *malaccensis*; **VII**, BB, ABB, AAB groups, *balbisiana*, ‘Pisang awak’, ‘Pelipita’, ‘Mysore’, ‘Kunan’, ‘Silk’; **VIII**, AA and AAA group, ‘Lacatan’, ‘Sucrier’, AA cv., ‘Indonesia I’; **IX**, AAA group, ‘Mutika’, ‘Lujugira’; **X**, AAA, AAB group, ‘Red’, ‘Indonesia’, AA cv., ‘Ambon’, ‘Pome’, ‘Gros Michel’; **XI**, AA group, *banksii*, *banksii* ‘sensu lato’, *banksii* derivatives; **XII**, ABB, AAB group, ‘Saba’, ‘Monthan’, ‘Maia Maoli’, ‘Popoulu’; **XIII**, AAB, ‘Iholena’, plantains. Group I served as an outgroup. Major clades and subclades are discriminated by color. The ‘Ma845’, ‘Ma846’, ‘Ma848′, ‘Ma850’, ‘Ma851’, ‘Ma852’ lines are clustered together in group VI and their positions are indicated by red arrows. (**C**) Local ancestry estimation for ‘Ma848’ and ‘Ma851’, using allele clustering against six ancestral diploid groups previously identified (Martin et al., 2020). g = group. g1 (black): *Musa balbisiana*; g2 (orange): *M. acuminata* ssp. *burmannica*/*siamea*; g3 (blue): *M. acuminata* ssp. *malaccensis*, g4 (green): *M. acuminata* ssp. *banksii*/*microcarpa*; g5 (purple): ‘AA’ cv. ‘Pisang Madu’. g6 (red): *M. acuminata* ssp. *zebrina*. Unknown (grey): unassigned. Homologous chromosomes are grouped together by a bracket and the number indicates the chromosome number. Chromosome length is indicated at a megabase (Mb) scale.

**Figure 2 pathogens-12-00289-f002:**
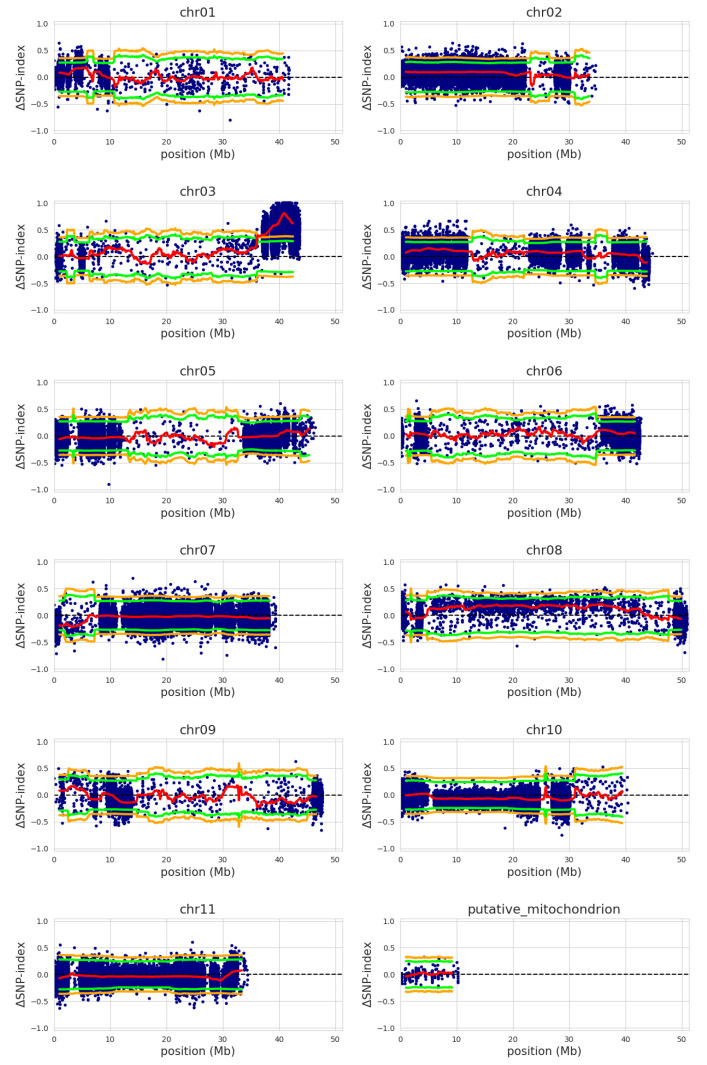
QTL-seq analysis using bulked F_2_ segregants. ΔSNP index between resistance (R) and susceptible (S) bulks were calculated and plotted on 11 *M. acuminata* ssp. *malaccensis* chromosomes and mitochondrion DNA. Mean ΔSNP indices are plotted (red). A window size of 2 Mb and a fixed step size of 100 Kb was used in plotting. Statistical confidence intervals under the null hypothesis of no QTLs detected are indicated at *p* < 0.05, green; *p* < 0.01, orange. Chromosome length and positions are indicated in Mb. The significant genomic region is at 36.2–42.4 Mb on chromosome 3.

**Table 1 pathogens-12-00289-t001:** Total number of variants including SNPs and InDels (Insertions and Deletions) detected on each chromosome in DH-Pahang v4.3.

Chromosome	Length (bp)	Variants	Variants Rate (Avg Length in bp/Variant)
1	41,765,374	291,420	143
2	34,826,099	286,153	121
3	43,931,233	315,282	139
4	45,086,258	347,648	129
5	46,513,039	360,938	128
6	43,117,521	341,605	126
7	39,373,400	296,922	132
8	51,314,288	362,193	141
9	47,719,527	366,392	130
10	40,511,255	227,914	177
11	34,663,808	241,783	143
Mitochondria	10,397,121	31,917	325
Total	479,218,923	3,470,167	138

## Data Availability

The data presented in this study are available on request from the corresponding author. The data are not publicly available due to confidentiality of genetic information pertaining gene discovery.

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
