# Peer review of "Identification of a Major QTL-Controlling Resistance to the Subtropical Race 4 of Fusarium oxysporum f. sp. cubense in Musa acuminata ssp. malaccensis"

_pathogens, 2023, doi:10.3390/pathogens12020289_

Round 1

Reviewer 1 Report

This manuscript characterized Foc Sub-tropical Race 4 (STR4) resistance in banana, and identified a major QTL for the resistance. There are some comments for this study.

1. In the candidate genes listed in supplementary Table2,  how to screen target gene responsible for the STR4 resistance in your future study? This reply can be for discussion.

2. Should the journal names in ref29 and ref30 be complete spelled or abbreviated? Please recheck the reference format.

3. The name of PNAS in ref48 and ref60 are different. Please double-check.

4. In Line556, "Doi" or "doi" ? Please check the full text. 

Author Response

Reviewer 1

Comments and Suggestions for Authors

This manuscript characterized Foc Sub-tropical Race 4 (STR4) resistance in banana, and identified a major QTL for the resistance. There are some comments for this study.

  1. In the candidate genes listed in supplementary Table2,  how to screen target gene responsible for the STR4 resistance in your future study? This reply can be for discussion.

Response: added the following sentences in the discussion (line 375-381) to address the future directions of this project, including how to screen for target candidate genes.

“Future work is currently being undertaken to fine map the QTL. Cleaved and amplified polymorphic sequence (CAPS) markers are being developed to saturate the candidate region. A linkage map is being developed. F2individuals carrying recombination events are being identified. Recombinants will be phenotyped with both Foc-STR4 and Foc-TR4 to validate and delimit the candidate region. Potential functional SNPs in R gene candidates will be converted into markers to test for co-segregation with the race 4 resistance controlled by this locus.”

  1. Should the journal names in ref29 and ref30 be complete spelled or abbreviated? Please recheck the reference format.

Response: All references have been re-checked and re-formatted according to the journal’s requirements.

  1. The name of PNAS in ref48 and ref60 are different. Please double-check.

Response: Those references have been corrected.

  1. In Line556, "Doi" or "doi" ? Please check the full text. 

Response: This has been corrected.

We would like to kindly thank the reviewer for taking his/her time to review this manuscript.

Reviewer 2 Report

I was pleased to read this report. Congratulations on your advancement in the field! I am not intimately familiar with the F. oxysporum f. sp. cubense, and consequently may not be able to detect all problems in this manuscript. Nonetheless, I believe that this is a very good contribution of identifying a source of resistance and a 6.2 Mb region with a large number of potential resistance gene candidates. I would have liked to have seen additional information on the chromosome 3 region, (e.g.,  presumably it’s in a location on the chromosome that recombines readily), or a shorter region than 6.2 Mb, but I assume that what is here is sufficient. 

Two very minor comments...

L154 “approximately 3:1” should be accompanied by a goodness of fit statistic & P-value

Table 1. No units stated for “Variants rate”, (avg length, bp/variant)

Author Response

Reviewer 2

Comments and Suggestions for Authors

I was pleased to read this report. Congratulations on your advancement in the field! I am not intimately familiar with the F. oxysporum f. sp. cubense, and consequently may not be able to detect all problems in this manuscript. Nonetheless, I believe that this is a very good contribution of identifying a source of resistance and a 6.2 Mb region with a large number of potential resistance gene candidates. I would have liked to have seen additional information on the chromosome 3 region, (e.g.,  presumably it’s in a location on the chromosome that recombines readily), or a shorter region than 6.2 Mb, but I assume that what is here is sufficient.

Response: We’ve added additional information to address the reviewer’s concerns. Line 375-381.

“Cleaved and amplified polymorphic sequence (CAPS) markers are being developed to saturate the candidate region. A linkage map is being developed. F2 individuals carrying recombination events are being identified. Recombinants will be phenotyped with both Foc-STR4 and Foc-TR4 to validate and delimit the candidate region. Potential functional SNPs in R gene candidates will be converted into markers to test for co-segregation with the race 4 resistance controlled by this locus.”

Two very minor comments...

L154 “approximately 3:1” should be accompanied by a goodness of fit statistic & P-value

Response: Added the following. “Chi-square goodness of fit with χ2 = 0.056, p = 0.81, df = 1, α = 0.05”

Table 1. No units stated for “Variants rate”, (avg length, bp/variant)

Response: Added “Avg length in bp/variant”.

We would like to kindly thank the reviewer for taking his/her time to review this manuscript.